# Transcriptomic and Metabolomic Responses in Cotton Plant to *Apolygus lucorum* Infestation

**DOI:** 10.3390/insects13040391

**Published:** 2022-04-15

**Authors:** Han Chen, Honghua Su, Shuai Zhang, Tianxing Jing, Zhe Liu, Yizhong Yang

**Affiliations:** College of Horticulture and Plant Protection, Yangzhou University, Yangzhou 225007, China; levyashin276@163.com (H.C.); susugj@126.com (H.S.); shuaizhang@yzu.edu.cn (S.Z.); jingtx@yzu.edu.cn (T.J.); zhe_liu9208@163.com (Z.L.)

**Keywords:** cotton, induced resistance, *Apolygus lucorum*, jasmonic acid, condensed tannins

## Abstract

**Simple Summary:**

To elucidate the cotton plant’s defense mechanism against *Apolygus lucorum*, the transcriptomic and metabolomic differences in cotton plants after infestation with *A. lucorum* were investigated. The results showed that *A. lucorum* feeding elicits a rapid and strong defense response involved in many primary and secondary metabolic processes during the whole infestation process. The jasmonic acid pathway was shown to mediate the defense mechanisms in cotton plants, and the salicylic acid pathway was attenuated after damage from *A. lucorum*. The accumulation of condensed tannins was also promoted in cotton plants as a defense mechanism against *A. lucorum*. These results provided comprehensive insights into the defense system and will help us to discover unknown defensive genes and improve the integrated pest management of *A. lucorum*.

**Abstract:**

With the wide-scale adoption of transgenic *Bacillus thuringiensis* (Bt) cotton, *Apolygus lucorum* (Meyer-Dür) has become the most serious pest and has caused extensive yield loss in cotton production. However, little is known about the defense responses of cotton at the seedling stage to *A. lucorum* feeding. In this study, to elucidate the cotton defense mechanism, cotton leaves were damaged by *A. lucorum* for 0, 4, 12 and 24 h. The transcriptomic results showed that *A. lucorum* feeding elicits a rapid and strong defense response in gene expression during the whole infestation process in cotton plants. Further analysis revealed that at each assessment time, more differentially expressed genes were up-regulated than down-regulated. The integrated analysis of transcriptomic and metabolic data showed that most of the genes involved in jasmonic acid (JA) biosynthesis were initially up-regulated, and this trend continued during an infestation. Meanwhile, the content levels of JA and its intermediate products were also significantly increased throughout the whole infestation process. The similar trend was displayed in condensed tannins biosynthesis. This research proved that, after plants are damaged by *A. lucorum*, the JA pathway mediates the defense mechanisms in cotton plants by promoting the accumulation of condensed tannins as a defense mechanism against *A. lucorum*. These results will help us to discover unknown defensive genes and improve the integrated pest management of *A. lucorum*.

## 1. Introduction

Plants can perceive several mechanical and chemical cues from pests, including feeding damage, oviposition and the presence of pests on their surfaces [1,2,3]. In addition, plants can activate defensive mechanisms to improve their fitness, divert resources away from plant growth and reduce major cell processes concerning photosynthesis or growth [4,5,6]. The defense system of a plant is divided into “direct defense” and “indirect defense”, depending on the mechanisms enacted. The former involves the biosynthesis of toxic compounds, which can suppress herbivore performance, while the latter is involved in the emission of herbivore-induced plant volatiles (HIPVs), which can attract natural enemies [7,8].

Previous studies have proved that these types of defenses are mediated by a series of phytohormones such as jasmonic acid (JA), salicylic acid (SA) and abscisic acid (ABA) [9,10,11,12]. In particular, JA and SA, along with their derivatives, play a primary role in modulating plant defense against complex biotic and abiotic environments, respectively [4].

Cotton (*Gossypium hirsutum* L.) is grown globally and used in textiles fabrics, fodder and other industries. During its lifetime, a cotton plant will unavoidably be subject to attacks from herbivores. Cotton plants also produce a range of compounds, including condensed tannins, gossypol, flavonoids and others, to reduce the effects of herbivore attacks. For example, after plants were damaged by *Spodoptera littoralis* (Boisd.), the content level of condensed tannins in cotton (var. Delta Pineland 90) leaves was shown to significantly increase [13], and the gossypol concentration was shown to increase by more than a fold in the days following cotton (var. Deltapine acala 90) plants being damaged by *Helicoverpa armigera* (Hübner) [14]. All of these defensive mechanisms work together in order for cotton plants to resist pest attacks.

Since transgenic *Bacillus thuringiensis* (Bt) cotton was adopted in the 1990s, *Apolygus lucorum* (Meyer-Dür), which historically has been a pest with secondary importance, has recently emerged as the most serious pest to cotton and has caused extensive yield loss in cotton production in China [15]. Although *A. lucorum* has become the dominant pest in cotton, how Bt cotton responds to *A. lucorum* at the seedling stage is still unclear.

In this study, to evaluate the dynamic defense responses of cotton to *A. lucorum* feeding, we investigated the responses of cotton plants to *A. lucorum* damage by integrating the results from cotton transcriptional dynamics with phytohormones and metabolite profiling experiments. These results will allow us to expound the molecular and biochemical defense mechanisms of cotton plants against *A. lucorum*.

## 2. Materials and Methods

### 2.1. Plant Growth

Bt cotton seeds (var. SGK 321) were provided by the Chinese Academy of Agricultural Sciences (CAAS). This cultivar has median-level resistance to *A. lucorum* and is widely grown near the Yangtze River. The cotton plants were grown in a greenhouse at Yangzhou University. Each plant was sown in an individual plastic pot (height, 15 cm; diameter, 20 cm). The temperature was maintained at 26~30 °C, and the plants were watered every week. All plants were placed in nylon cages to prevent them being damaged by herbivores. The plants were spaced 2 m apart to avoid communication between each other. After 8 leaves emerged on each plant, healthy cotton plants were chosen for use in the experiments.

### 2.2. Insect Rearing

Before the start of the experiment, we established colonies of *A. lucorum* at Yangzhou University, starting with eggs provided by CAAS. *A. lucorum* was reared on maize (*Zea mays* L.) and French bean (*Phaselus vulgaris* L.) at 25 °C, with 65% RH and a 14 h light:10 h dark photoperiod [16].

### 2.3. Plant Treatments

After being starved for 8 h, 4 third-stage nymphs were placed on top of each cotton plant and allowed to feed ad libitum in a nylon cage. Leaves were harvested for analysis at 0, 4, 12 and 24 h after the initial *A. lucorum* infestation. All plant material was collected in the early morning at the same time on the same day. All harvested samples were immediately frozen using liquid nitrogen for more than one hour and then stored in a −80 °C freezer. Three samples were prepared for RNA-seq data analysis, eight samples for metabolomics and six samples for phytohormones and condensed tannins analysis.

### 2.4. Total RNA Extraction, Transcriptome Sequencing, RNA-seq Data Analysis and qRT-PCR Analysis

Total RNA from each sample was isolated following the CTAB (Cetyltrimethylammonium Bromide) method, while the concentration and quality were assessed using an Agilent 2100 Bioanalyzer (Agilent, Palo Alto, CA, USA) [17]. Clean reads were mapped to *Gossypium hirsutum* v2.1 genome(accessed on 5 May 2020) downloaded from the NCBI database (https://www.ncbi.nlm.nih.gov) using HISAT2 (v2.1.0), and only unique mapping reads were used to calculate gene expression [18]. The transcriptional assembly was performed using Bowtie software (v2.25), and fragments per kilobase of exon per million fragments (FPKM) was used to calculate gene expression levels with RSEM software (v1.2.8). Differentially expressed genes (DEGs) were identified with an R package with |log_2_ Fold Change| ≥ 1 and adjusted *p*-value ≤ 0.001 [19]. The Kyoto Encyclopedia of Genes and Genomes (KEGG) enrichment analyses of the DEGs were performed using the cluster Profiler R package with padj <0.05 to find significant enrichments. To verify the RNA-seq data, qRT-PCR analysis was performed using iTaqTM Universal SYBR^®^ Green Supermix (BIO-RAD, Berkeley, CA, USA). To calculate expression changes, the relative 2^−ΔΔCt^-method was used, and the *Histone*-3 (gene ID: 107950429) gene was used as a candidate reference gene [20,21]. All primers that were used are listed in Appendix A.

### 2.5. Untargeted Metabolite Analysis

For metabolite analysis assays with regard to the cotton, 25 mg of leaf tissue was weighed, and 800 μL of extraction solvent (methanol/water, 1:1, *v*/*v*) was added to every sample [22]. Then, the homogenized solution was centrifuged at 30,000× *g* and 4 °C for 20 min, and 550 μL of supernatant was transferred into a new tube. The analysis was performed using a UPLC (Waters, Milford, MA, USA) system with an ACQUITY UPLC T3 column (100 mm × 2.1 mm, 1.8 μm, Waters, Milford, MA, USA), and the chromatographic condition was identical to that used by Peng [23]. To detect metabolites in each sample, a high-resolution tandem mass spectrometer (Xevo G2 XS, Q-TOF-MS, Waters, Manchester, UK) was used in both positive and negative ion modes. The parameters were also the same as those used by Peng [23]. For qualitative and quantitative analyses, all raw data were analyzed using Progenesis QI (version 2.2) and the online KEGG database. To determine differentially accumulated metabolites (DAMs) between cotton damaged by *A. lucorum* and healthy cotton, the orthogonal projections to latent structure discriminant analysis (OPLS-DA) was used. The relative importance of each metabolite to the OPLS-DA model was checked using the variable importance in projection (VIP) metric [24]. Metabolites with VIP ≥ 1.0, fold change > 1.2 or fold change < 0.83 and *p*-value < 0.05 were considered as DAMs for group comparations.

### 2.6. Content of Phytohormones and Condensed Tannins Analysis

Deep-frozen cotton leaves were lyophilized. The quantitation of JA, SA, ABA and JA-isoleucine conjugate (JA-Ile) were analyzed using a coupled liquid chromatography–mass spectrometry system (LC-MS/MS, Waters, Milford, MA, USA) following Sun [25]. Condensed tannins were extracted and measured following Lin [26].

### 2.7. Statistical Analysis

Statistical comparisons of phytohormones, condensed tannins and their intermediate products from different treatments were all carried out by one way-ANOVA followed by the Tukey’s HSD test using the software SPSS 25, and a significance level of *p* < 0.05 was applied.

## 3. Results

### 3.1. RNA-seq Analysis of Cotton Response to A. lucorum Attack

To investigate global transcriptomic changes in response to *A. lucorum* feeding, data from leaves damaged by *A. lucorum* for 0, 4, 12 and 24 h were collected. In total, 56,242 genes were detected from all samples (Appendix A). After the initiation of *A. lucorum* feeding, 13,254 genes showed altered expression at each time point. The samples at 4, 12 and 24 h exhibited 4623 (3847 up-regulated and 776 down-regulated), 6109 (4374 up-regulated and 1735 down-regulated) and 10,851 (6236 up-regulated and 4615 down-regulated) DEGs, respectively (Figure 1a). The number of up-regulated genes was superior to down-regulated DEGs at each time point assessed, while the number of down-regulated DEGs increased much more rapidly than the number of up-regulated DEGs (Appendix A). The distribution of up-regulated and down-regulated DEGs was calculated, and this information is presented in a Venn diagram (Figure 1b,c). A large number of genes (a total of 7964) increased in prevalence, but the number of genes (a total of 2262), which increased at every time point, was severely limited. In addition, a unique set of genes decreased (a total of 5399), while only 249 genes decreased at every time point. A total of 1117 DEGs (820 up-regulated and 297 down-regulated) only occurred 4 h post-infestation and then returned to normal, while 3103 DEGs (2304 up-regulated and 3235 down-regulated) occurred 24 h post-infestation.

To further verify the RNA-seq results, eight DEGs were selected for qPCR analysis. The expression patterns of the DEGs obtained from qPCR were consistent with those from RNA-seq (Appendix A).

### 3.2. KEGG Analysis of Cotton Response to A. lucorum Attack

Subsequently, all DEGs were subjected to KEGG pathway analysis. The DEGs at 4, 12 and 24 h after *A. lucorum* feeding were classified into 10, 15 and 16 significant KEGG pathways, respectively (*p* < 0.05) (Appendix A). In these significant KEGG pathways, the biosynthesis of phenylpropanoid, flavonoid and isoflavonoid compounds as well as the metabolism of α-linolenic acid, plant hormone signal transduction, MAPK signaling and plant–pathogen interaction were involved in the responses of cotton to *A. lucorum* at each time point. These KEGG pathways were considered to be involved in the defense mechanisms of cotton in response to herbivores.

### 3.3. Dynamic Transcriptome Responses to A. lucorum Feeding

To better understand the dynamics of cotton transcriptomes in response to *A. lucorum* herbivory, the total DEGs were subjected to different clusters using Short Time-series Expression Miner analysis (STEM) [27]. The DEGs were clustered into 13 important temporal gene expression profiles (Figure 2). Profile 40 included 1015 transcripts that showed a trend of up-regulated expression during the whole 24 h after *A. lucorum* feeding, and transcripts in profile 42 and 48 also showed this trend. Transcripts in the abovementioned profiles were immediately up-regulated at 4 h and continued to be up-regulated until 24 h. The transcripts in profile 45, 46 and 49 also showed a trend of up-regulated expression in the whole 24 h, but they reached a peak at 4 h or 12 h. Pathway enrichment analysis showed that the transcripts in the two profiles mentioned above were mainly related to the biosynthesis of phenylpropanoids, α-linolenic acid, flavonoids and pathways of plant hormone signal transduction, MAPK signaling and plant–pathogen interaction (Appendix A). This result suggested the defense response of cotton was immediately induced by *A. lucorum* herbivory and was continuously induced for a long time. Profile 1, 9 and 11 showed a trend of down-regulated expression, and most genes included in these profiles were involved in the primary metabolism, including C5-branched dibasic acid metabolism, glucosinolate and photosynthesis-antenna proteins biosynthesis. These up- and down-regulations in genes involved in cotton’s primary metabolism might be caused by the readjustment of the primary metabolism in response to pest attacks.

### 3.4. Metabolomic Analysis of Cotton Responses to A. lucorum Attack

To evaluate metabolic changes in response to *A. lucorum* infestation, cotton leaves were analyzed using UPLC-QTOF-MS in both the positive and negative ion mode. Metabolite abundance increased more slowly than gene expression. In total, 4710 metabolites were detected from all samples (Appendix A). There were 266 (106 up-regulated and 160 down-regulated), 721 (422 up-regulated and 299 down-regulated) and 1338 (780 up-regulated and 558 down-regulated) DAMs in the samples at the 4, 12 and 24 h time points, respectively (Figure 1d). The number of up-regulated DAMs was only lower than the number of down-regulated DAMs at 4 h, while the number of up-regulated DAMs was higher at later time points. As shown in Figure 1, only a few metabolites (10 up-regulated and 32 down-regulated) changed at each time point (Figure 1e,f).

### 3.5. Phytohormone Levels Induced by A. lucorum

To evaluate the phytohormone level changes induced by *A. lucorum*, the concentrations of JA, JA-Ile, SA and ABA in cotton leaves were analyzed. After plants were damaged by *A. lucorum*, the JA content level continuously increased, and the content level at 24 h was significantly higher than those at other time points. The content level at 12 h was also significantly higher than that at the initial time point. The JA-Ile level showed a similar trend, but only increased significantly at 24 h. The level of SA decreased significantly during the whole infestation process, while the level of ABA did not change significantly at any time point (Figure 3).

### 3.6. Biosynthesis of Jasmonic Acid

In cotton plants, jasmonic acid is biosynthesized from α-linolenic acid by rate-limiting enzyme allene oxide synthase (AOS) and other crucial enzymes, including lipoxygenase (LOX), allene oxide cyclase (AOC), 12-oxophytodienoate reductase (OPR), OPC8-CoA ligase (OPCL), acyl-CoA oxidase (ACX), enoyl-CoA hydratase (MFP) and acetyl-CoA acyltransferase (fad A) [28,29]. To explore the role of the JA pathway in response to *A. lucorum* feeding, the expression levels of genes involved in JA biosynthesis were analyzed, and the concentrations of intermediate products involved in JA biosynthesis were measured (Figure 4). As shown in Figure 4b, most of the genes that encode the rate-limiting enzyme AOS and other crucial enzymes, including LOX, AOC, OPR, OPCL, ACX, MFP and fad A, were all up-regulated at the initial time point and continued to be up-regulated during the infestation. Further chemical analysis revealed that the precursor α-linolenic acid was significantly decreased at the 4 h time point and remained at a low level during the infestation, and the content level of (9Z,11E,15Z)-(13S)-13-hydroperoxyoctadeca-9,11,15-trienoic acid (13(S)-HPOT) did not significantly change during the whole infestation process. However, the levels of other upstream products, including (9Z,15Z)-(13S)-12,13-Epoxyoctadeca-9,11,15-trienoic acid (12,13(S)-EOT) 12-Oxo-10,15(Z)-phytodienoic acid (12-OPDA), were both significantly increased at the 24 h time point (Figure 4c). The content levels of 7-Isojasmonic acid CoA at 12 h and 24 h were both significantly higher than those at other time points. Due to the content levels of JA and JA-Ile being significantly increased, we determined that the JA pathway was intensely induced by *A. lucorum* feeding.

### 3.7. Biosynthesis of Condensed Tannins

In cotton plants, condensed tannins have protective roles against insect attacks [30]. They were biosynthesized from *L*-phenylalanine using a series of crucial enzymes including *L*-phenylalanine ammonialyase (PAL), cinnamate 4-hydroxylase (C4H), 4-coumarate-CoA ligase (4CL), chalcone synthase (CHS), chalcone isomerase (CHI), flavanone 3-hydroxylase (F3H), flavonoid 3′-hydroxylase(F3′H), dihydroflavonol reductase (DFR), anthocyanidin synthase (ANS) and anthocyanidin reductase (ANR) [31]. In our study, the expression of genes and the concentrations of intermediate products involved in condensed tannins biosynthesis were analyzed (Figure 5). As shown in Figure 5b, nearly all of the genes that encode crucial enzymes were activated in turn. The content levels of *L*-phenylalanine and p-Coumaric acid were significantly increased at 4 h and then returned to normal. The trans-Cinnamic acid level was significantly higher at 4 h than those at other time points, but the content levels at 12 h or 24 h were also significantly higher than the initial level. After plants were damaged by *A. lucorum*, the level of Dihydrokaempferol was significantly increased during the whole infestation process, and the level of condensed tannins was also found to be significantly increased after 24 h. These results indicate that the condensed tannins pathway was also intensely induced by *A. lucorum* feeding.

## 4. Discussion

Since the commercialization of Bt cotton, *A. lucorum* has gradually become the most significant pest in cotton-growing regions in China [15]. However, knowledge of defense mechanisms in response to *A. lucorum* infestation has remained limited. In our study, we analyzed the dynamics of transcriptomes in Bt cotton SGK321 after being attacked by *A. lucorum*. Our analysis revealed that after plants were damaged by *A. lucorum*, the gene expression in leaves changed continually in next 24 h, and more DEGs were up-regulated than those that were down-regulated at each time point. This result indicated that *A. lucorum* feeding induces more induction than suppression in gene expression. Similar results were found after the evaluation of the transcriptomic changes in cotton plants after chewing insects, such as *H. armigera* or *Anthonomus grandis* (Bohemian), had fed on them [32,33]. However, when cotton plants were damaged by piercing–sucking insects such as *Bemisia tabaci* (Gennadius) or *Aphis gossypii* (Glover), more DEGs were down-regulated than those that were up-regulated at each time point [34,35]. The difference in global transcriptome changes might be caused by the feeding behavior of *A. lucorum*. Unlike other vascular feeders, such as aphids and whiteflies, *A. lucorum* is a mesophyll feeder, which uses its stylet to probe into plant host tissue and shred cells in order to feed on the tissue [36]. Due to this “lacerate and flush” feeding strategy, *A. lucorum* causes more physical damage to cells than aphids or whiteflies, and the damage they cause is similar to that caused by chewing insects [37].

Plant-induced defense to herbivores is mediated by phytohormones, especially JA and SA, which both play fundamental roles in the modulation of plant defense against herbivores [38]. SA-JA crosstalk in plant–herbivore interactions determined the effectiveness of induced resistance [39]. In our study, genes involved in the KEGG pathway of “α-linolenic acid metabolism” were activated after plants were damaged by *A. lucorum*. The increase in contents of JA and its intermediate products was also detected at the time points analyzed. Meanwhile, the expression of crucial enzyme isochorismate synthase (ICS) in SA biosynthesis was reduced, and the content of SA decreased significantly during the whole infestation process (Appendix A). These results indicated that the JA pathway plays a primary role in cotton’s defense mechanisms against *A. lucorum*, and the SA pathway is attenuated after cotton plants are damaged by *A. lucorum.* However, a previous study considered SA signaling to be dominant in cotton defense mechanisms against *A. lucorum*, and in another study, it was suggested that the JA pathway is responsible for the elicitation of cotton defense mechanisms against *A. lucorum* without attenuating the SA pathway [40,41]. Another study proved that different genotypes of plants will show totally different induced defense mechanisms [42]. Some research also proved that plant-induced defense mechanisms follow a pest density-dependent response [43,44]. We speculated that the differences in gene expression in both JA and SA may be induced by the different densities of pests or different cultivars of cotton. This work should be completed in future.

Condensed tannins were the most important secondary metabolite in cotton’s defense mechanisms against pests. Some previous studies have proved that condensed tannins could minimize the damage caused by *A. lucorum*, and their content levels had a significant positive correlation with cotton’s resistance to *A. lucorum* [26,45]. Recently, studies have explained resistance mechanisms in which condensed tannins had a negative impact on probing the behaviors of *A. lucorum* [46,47]. In our study, most of the genes that encode crucial enzymes in condensed tannins biosynthesis were up-regulated during the whole infestation process. The content levels of condensed tannins and their intermediate products were also significantly increased after plants were damaged by *A. lucorum* for 24 h. This phenomenon indicated that cotton plants activate a defense system to resist *A. lucorum* damage.

However, condensed tannins only provided an incomplete picture of the direct defense responses of cotton plants to pests, and another defense substance, gossypol, also inhibited insect performance. Previous studies have proved gossypol also had a negative impact on the probing behaviors of *A. lucorum* [47]. In our study, we found that the content level of (+)-δ-cadinene, which has always been considered to be an initial material of gossypol, was significantly increased in herbivore-induced plant volatiles, while the content level of gossypol in leaves was significantly decreased (unpublished). A recent study suggested that cotton plants activate a specific and mitigated defense strategy to *A. lucorum* feeding. This defense strategy replaces the gossypol-related defense mechanism because less extensive damage is caused by the stylets of *A. lucorum* [41]. Although the gossypol biosynthetic pathway has been reported [48], in our study, the genes that encode crucial enzymes and contents of intermediate products were not analyzed. This work should be completed in future.

## 5. Conclusions

In our paper, we evaluated the induced defense of Bt cotton at the seedling stage in response to *A. lucorum* attacks. The dynamic transcriptomic analysis showed a rapid and strong response in the first 4 h after being attacked by *A. lucorum*, which continued to cause changes until 24 h had passed. The integrated analysis of the transcriptomic and metabolic data revealed that JA mediates the defense mechanisms in cotton plants, and condensed tannins manage the defense mechanism against *A. lucorum*. This work not only clarified the defense mechanism of cotton against *A. lucorum*, but might be of significance in the discovery of unknown defensive genes, the breeding of cotton cultivars and improving the integrated pest management of *A. lucorum*.

## Figures and Tables

**Figure 1 insects-13-00391-f001:**
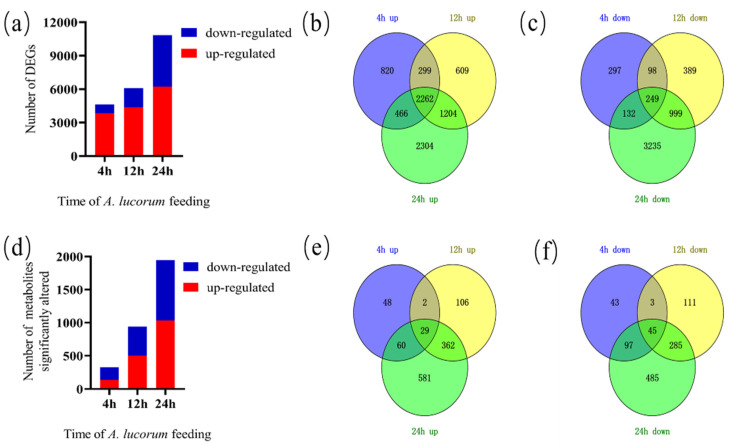
Transcriptomic and metabolomic overview of a time course of *A. lucorum* damage on cotton plants. (**a**) Total number of individual transcripts that were significantly up- or down-regulated at each time point. (**b**) Venn diagram illustrating the number of genes up-regulated during the time course. (**c**) Venn diagram illustrating the number of genes down-regulated during the time course. (**d**) Total number of metabolites that were significantly up- or down-regulated at each time point. (**e**) Venn diagram illustrating the number of metabolites up-regulated during the time course. (**f**) Venn diagram illustrating the number of metabolites down-regulated during the time course.

**Figure 2 insects-13-00391-f002:**
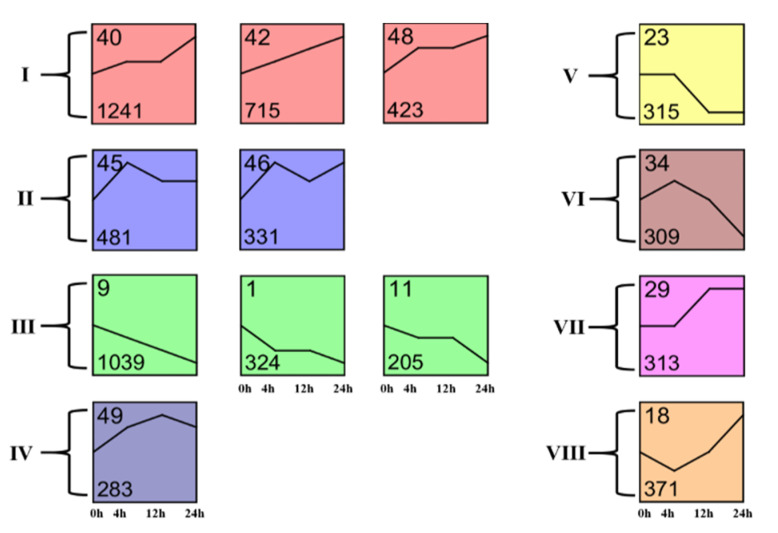
Clustering and classification of differentially expressed genes. The Roman numerals on the left indicate the class. The number in the top left corner in each panel indicates the identification number (ID), and the number in the bottom left corner of each panel indicates the number of genes in the cluster.

**Figure 3 insects-13-00391-f003:**
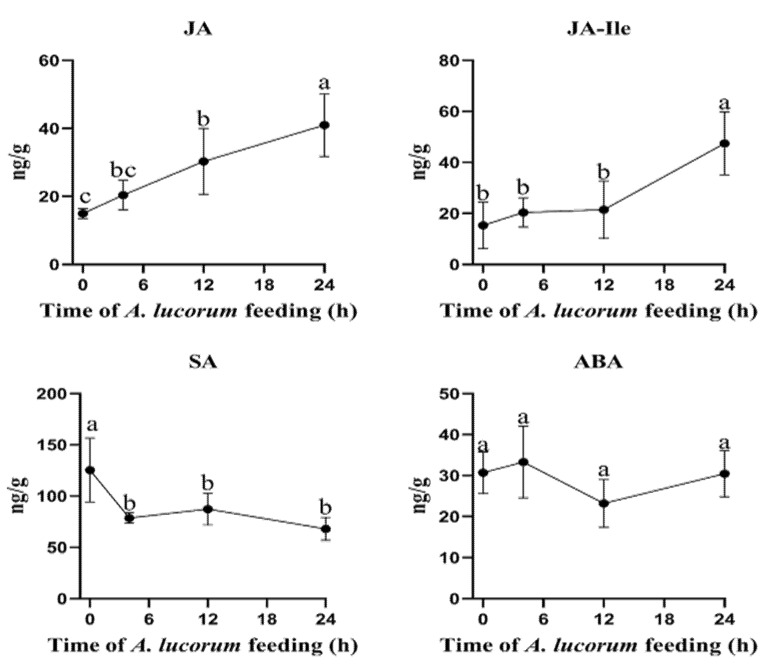
Plant phytohormones changed after *A. lucorum* feeding on cotton leaves. JA, jasmonic acid; JA-Ile, jasmonoyl-L-isoleucine; ABA, abscisic acid; SA, salicylic acid. Mean ± SE of *n* = 6; amounts with different letters were significantly different at the 5% level according to Tukey’s HSD test.

**Figure 4 insects-13-00391-f004:**
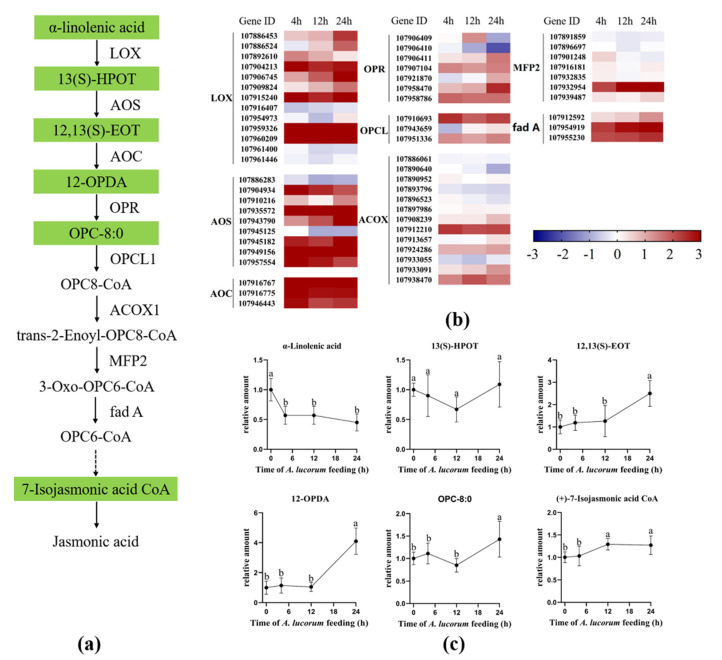
*A. lucorum*-induced responses in the jasmonic acid (JA) pathway. (**a**) Overview of the JA pathway. Metabolites shaded in green were measured. Solid arrows represent established biosynthesis steps, while broken arrows indicate the involvement of multiple enzymatic reactions. (**b**) Heat map of the expression of genes associated with the JA pathway. (**c**) Intermediate products contents in JA pathway cotton leaves. Values are mean ± SE of six biological replicates. Amounts with different letters were significantly different at the 5% level according to Tukey’s HSD test. Lipoxygenase (LOX); (9Z,11E,15Z)-(13S)-13-hydroperoxyoctadeca-9,11,15-rienoic acid (13(S)-HPOT); allene oxide synthase (AOS); (9Z,15Z)-(13S)-12,13-epoxyoctadeca-9,11,15-trienoic acid (12,13(S)-EOT); allene oxide cyclase (AOC); 12-oxo-10,15(Z)-phytodienoic acid (12-OPDA); 12-oxophytodienoate reductase (OPR); 8-[(1R,2R)-3-Oxo-2-{(Z)-pent-2-enyl}cyclopentyl]octanoate (OPC-8:0); OPC8-CoA ligase (OPCL); acyl-CoA oxidase (ACX); enoyl-CoA hydratase (MFP); acetyl-CoA acyl-transferase (fad A).

**Figure 5 insects-13-00391-f005:**
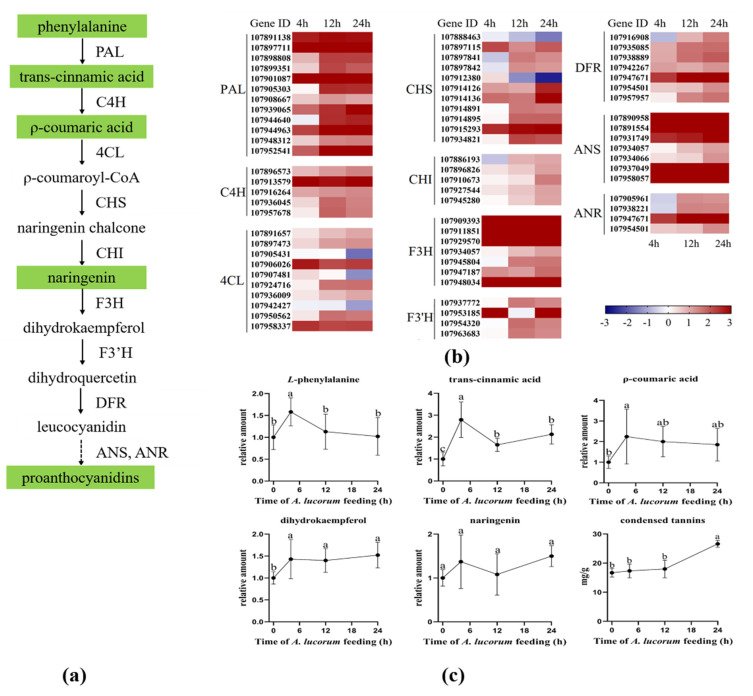
*A. lucorum*-induced responses in condensed tannins pathway. (**a**) Overview of the condensed tannins pathway. Metabolites shaded in green were measured. Solid arrows represent established biosynthesis steps, while broken arrows indicate the involvement of multiple enzymatic reactions. (**b**) Heat map of the expression of genes associated with the condensed tannins pathway. (**c**) Intermediate products contents in condensed tannins pathway cotton leaves. Values are mean ± SE of six biological replicates. Amounts with different letters were significantly different at the 5% level according to Tukey’s HSD test. *L*-phenylalanine ammonialyase (PAL), cinnamate 4−hydroxylase (C4H), 4−coumarate: CoA ligase (4CL), chalcone synthase (CHS), chalcone isomerase (CHI), flavanone-3-hydroxylase (F3H), flavonoid 3′−hydroxylase (F3′H), dihydroflavonol reductase (DFR), anthocyanidin synthase (ANS), anthocyanidin reductase (ANR).

## Data Availability

The data presented in this study are available in article and Appendix A.

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
