# Peer review of "Transcriptomic and Metabolomic Responses in Cotton Plant to Apolygus lucorum Infestation"

_insects, 2022, doi:10.3390/insects13040391_

Round 1

Reviewer 1 Report

General comments:

The manuscript needs a linguistic revision to achieve a more fluent style.

Avoid conditionel tense in your text, such as "Cotton plants would also produce ..." (p. 2, lines 50-51). Plants really do so, at least some plants. Better be explicit and give these plants as example. The response to herbivore attack differs between plant species and often also between cultivars/genotypes within a plant species. Do not generalize the responses for ALL plants when found, for example, in Arabidopsis or few other plant species. Better name the species where the responses you refer to really have been proved. You may then postulate that the same response may be found in the cotton cv. of your work.

Specific comments besides those given in the attached commented pdf of the ms.:

p. 1and 2: Besides the following points, the complete introduction requires thorough linguistic revision.

p. 1, line 26-27: These sentences should be revised as their meaning is not consistent with the statements of the literature cited. Perhaps, due to linguistic deficiencies the meaning of the sentences is unclear. Be more explicit and refer to the biological system given in the cited literature. Where the authors of the literature cited refer themselves to literature cited, take these articles to refer to.

p. 2, line 70: Please describe the growing conditions of your plants in detail (soil/pot ground type, potting trays, climatic conditions, place

p. 2, line 83: "freely"? do you mean "ad libitum"?

p. 3, lines 111-115: cut sentences shorter, merge informations.

p. 3, lines 132-135: please give the statistical analysis in detail. Were the data tested for normal distribution, if not, were they transformed, provide the variance analysis data.

p. 3-4, lines 147-149: rephrase sentence

p. 4, line 150: be precise why the expression was "of particular note"? Rephrase also the term "of particular note".

p. 4, lines 159-163: split this sentence in two

p. 7, lines 255-261: condense these sentences

p. 7-9: the discussion must completely be linguistically revised. Some expressions are unclear.

p. 9, lines 338-343: don't repeat what you have done in the conclusions but tell what the findings are good for.

Reviewer 2 Report

The manuscript Transciptomic and metabolomic responses in cotton plant to Apolygus lucorum infestation reports on the infestation of 8 leave stage cotton plants that were infested with 4 third-stage-old nymphs. RNA-seq and metabolomics data are reported and the RNA seq data for some genes are supported with RT-qPCR data.

Is the cultivar SGK321 used in the experiments susceptible, tolerant or resistant to A. lucorum? Why was this cultivar used?

The Methods section can be improved as it is not clear from the experimental layout or the statistical analyses what is eventually reported in the Results where treated plants are compared to controls. Are the data at each timepoint compared to the corresponding control timepoint or the time 0 timepoint? From the experimental layout it is not clear if there were non-infested plants that were kept and harvested at each of the later timepoints or not. This is especially important for figures 4, 5c and 6c.

The qPCR layout indicates that only one reference gene, Histone-3, was used for normalising data against. The accepted standard for normalisation of qPCR data is to use at least to reference genes.

The Results presented in Figure 2 lacks statistical analyses to indicate if the RNA seq data is significantly different or not to the qPCR data. 

In section 3.5 there is an error in the interpretation of the data presented in Figure 5C. The manuscript states that 'while other upstream products 235 including (9Z,15Z)-(13S)-12,13-Epoxyoctadeca-9,11,15-trienoic acid(12,13(S)-EOT), 12-236 Oxo-10,15(Z)-phytodienoic acid (12-OPDA), and 7-Isojasmonic acid CoA were all accu-237 mulated throughout the whole infestation and significantly increased at 24h time point (Fig 5c)'. The data presented in this figure shows that only Iso-Jasmonic acid CoA is upregulated at 12 and 24 hpi. All the other tested intermediates are either significantly downregulated (13(S)-HPOT) or not significantly regulated at all. The only significant regulation is at 24 hpi. The way that the data are presented is also lacking statistical analysis for the regulation within the infested plant. The plants are not statistically significant from the controls, but are the later timepoints significantly different from each other in the treated plants? A rigorous statistical analyses of the data will add value to the results.

The manuscript mentions that the difference in SA results obtained here and those for other studies lies within the numbers of A. lucorum used. This is an unsubstantiated hypothesis as the two cited studies used 2 and 15 A. lucorum nymphs or adults compared with the 4 in this study. The differences in cultivars used and their responses to A. lucorum, plant age at time of experiment and the timepoints under study may reveal more about this discrepancy than the numbers of A. lucorum used in the experiment. Although this effect cannot be discounted, no evidence is presented in support for this.

Round 2

Reviewer 1 Report

The manuscript has been improved pretty well.